# FineFed: Forward-Only Federated Fine-Tuning for Many-Class Tasks under Non-IID Heterogeneity

## Abstract

Federated learning (FL) on resource-constrained edge devices faces significant challenges when training large transformer models, particularly due to memory and computational limitations. While parameter-efficient fine-tuning (PEFT) methods help reduce memory usage, they still require back-propagation for gradient computation, which often demands more memory than storing model parameters. Forward-gradient (zero-order) FL offers a promising alternative by eliminating back-propagation, but existing methods suffer from computational inefficiency, poor performance on many-class tasks, and unstable convergence under non-IID data distributions.

We present *FineFed*, an efficient forward-only FL framework that addresses these limitations through three key innovations: (i) **Forward-Only Head Tuning**, which enables exact gradient computation for many-class classification heads without back-propagation; (ii) **Uncertainty-Guided Forward Gradient Estimation**, which reduces computational cost by approximately $2.5\times$ via uncertainty-guided sample selection and micro-batch perturbations; and (iii) **Shared Momentum**, which ensures stable local updates and fast convergence under extreme non-IID data heterogeneity. Comprehensive evaluations across NLP and vision datasets demonstrate that FineFed achieves superior model accuracy and system efficiency compared to state-of-the-art methods, making forward-only federated learning practical for real-world deployment. Our code is available at `https://anonymous.4open.science/r/FineFed-0554/`.

## 1 Introduction

Federated learning (FL) (McMahan et al., 2017) enables decentralized devices to collaboratively train models without sharing local data. Early FL deployments predominantly used smaller-scale deep neural networks, but recent practice increasingly adopts transformer-based models (Vaswani et al., 2017; Wu et al., 2020) across language and vision. Training transformer-based models requires substantial memory and computation, and as model capacity grows, per-client requirements increase accordingly. In FL, this trend conflicts with the tight resource budgets of edge devices: limited memory and compute, together with concurrent on-device workloads, make local training of large transformer-based models costly or impractical in the real world.

Recent studies have focused on parameter-efficient fine-tuning (PEFT) (He et al., 2022; Han et al., 2024) and forward-gradient-based approaches (Baydin et al., 2022), both aiming to mitigate memory and compute bottlenecks in FL. Despite their advantages, PEFT methods still need to cache activations for gradient computation during back-propagation (LeCun et al., 2015; Werbos, 1990), which often requires more memory than needed for storing model parameters. This reliance limits their potential to further reduce memory consumption, motivating alternatives that avoid back-propagation altogether.

Alternatively, zeroth-order optimization (Baydin et al., 2022) explores training models without back-propagation by estimating gradients via *perturbed inferences*, thereby avoiding activation caching. Recent studies (Xu et al., 2024) further combine forward training with PEFT (e.g., Adapter (Pfeiffer et al., 2020), BitFit (Zaken et al., 2022)) to better accommodate memory-constrained FL clients.

**However, existing forward training frameworks remain computationally expensive and systematically limited in FL.** Achieving accurate forward gradients typically requires many perturbed inferences, increasing client compute. In practice, prior forward training frameworks for FL (e.g., FedZO (Fang et al., 2022), DeComFL (Li et al., 2024b), FwdLLM (Xu et al., 2024)) exhibit three salient limitations: (i) they are commonly constrained to tasks with few classes and degrade on many-class tasks, (ii) forward-gradient estimation demands numerous perturbations on resource-constrained edge devices, incurring high computation cost, and (iii) non-IID data heterogeneity is not explicitly handled, leading to unstable and infrequent local updates and thus convergence difficulties. Combined, naive forward-only training is challenging to deploy at scale in practice.

**To overcome this limitation, we propose *FineFed*, an efficient forward training framework for federated learning.** *FineFed* introduces three key innovations—forward-only head tuning, uncertainty-guided forward gradient estimation, and shared momentum—that together improve the efficiency of forward gradient computation and enhance convergence, all while preserving the memory-saving benefits of forward training and parameter-efficient fine-tuning (PEFT).

**Contributions.** We make the following contributions:

- **Forward-only head tuning for many-class adaptation.** We obtain exact head gradients via forward-only manual computation, directly addresses many-class head optimization challenge while keeping entire training process forward-only.
- **Uncertainty-guided forward gradients with micro-batch perturbations.** We select top-uncertainty samples and perform micro-batch perturbations to concentrate computation on informative data, reducing computation cost for gradient estimation by $\approx \mathbf{2.5\times}$ while achieving comparable accuracy (Table 3).
- **Shared momentum for stable local updates under non-IID.** We broadcast shared momentum alongside parameters so clients can perform frequent and stable local updates, improving rounds-to-target and final accuracy versus removing momentum (Table 4).
- **Comprehensive evaluation and system analysis.** Extensive experiments across NLP and vision datasets, with clear compute/memory/communication accounting and thorough ablations that quantify each component's effectiveness (Sec. 4).

The remainder of this paper is organized as follows: Section 2 reviews background and motivates our design; Section 3 details the *FineFed* methodology; Section 4 presents experiments and system analysis. Additional implementation details, extended results, and ablations are provided in the Appendix.

## 2 BACKGROUND AND MOTIVATION

### 2.1 FORWARD GRADIENT COMPUTATION

Parameter-Efficient Fine-Tuning (PEFT) methods (He et al., 2022; Han et al., 2024) reduce the number of trainable parameters in FL, but still require caching intermediate activations for back-propagation, which can exceed available memory on edge devices. This limitation motivates alternatives that avoid back-propagation altogether.

Instead of computing gradients through back-propagation, forward training methods estimate gradients by zeroth-order optimization. Specifically, they compute directional derivatives through a forward pass with randomly perturbed parameters (Baydin et al., 2022). The directional derivative of a function $f$ with respect to a vector $v$ at a point $\theta$ is defined as (Xu et al., 2024; LeCun et al., 2015; Rumelhart et al., 1988):

$$\nabla_v f(\theta) := \lim_{h \to 0} \frac{f(\theta + hv) - f(\theta)}{h}, \tag{1}$$

where $v \sim \mathcal{N}(0, I)$ represents a perturbation direction and $\theta$ denotes the parameter vector. The forward gradient is estimated through the directional derivative $\nabla_v f(\theta)$:

$$g_v(\theta) := \nabla_v f(\theta)\, v := \big(\nabla f(\theta) \cdot v\big)\, v, \tag{2}$$

where $g_v(\theta)$ is an (asymptotically) unbiased estimator of the gradient $\nabla f(\theta)$ (Baydin et al., 2022). This allows forward gradients to be computed solely through a forward pass, referred to as *perturbed*

*inference*. In practice, one may approximate $\nabla_v f(\theta)$ using either a one-sided difference, $\big(f(\theta + hv) - f(\theta)\big)/h$ (used by DeComFL), or a symmetric difference, $\big(f(\theta + hv) - f(\theta - hv)\big)/(2h)$ (used by FwdLLM). Although the symmetric form reduces bias and often variance, it doubles forward passes per perturbation; given the limited computation resources on edge devices, we adopt the one-sided estimate in FineFed.

A key insight for FL is that the perturbation vector $v$ can be generated from a random seed, and the directional derivative $\nabla_v f(\theta)$ is a scalar value. This enables communication-efficient FL by exchanging only seeds and scalar projections between clients and server, rather than full model parameters or gradients. Existing works like FedZO (Fang et al., 2022) and DeComFL (Li et al., 2024b) leverage this mechanism to achieve KB-level communication through seed/projection exchange, enabling model parameter reconstruction and aggregation without transmitting complete gradients.

## 2.2 Existing Forward Training Frameworks in Federated Learning

Several existing works have integrated forward training methods into federated learning (FL) settings, each with distinct approaches and trade-offs. In this section, we introduce the two most representative works: FwdLLM (Xu et al., 2024) and DeComFL (Li et al., 2024b), and defer the discussion of other related works to Appendix K.

### 2.2.1 FwdLLM: Server-Side Accumulation with Low Update Frequency

**FwdLLM** proposes a forward training framework that combines forward gradient estimation with PEFT to reduce local memory and computation costs. To improve training stability, FwdLLM employs server-side variance control where the server accumulates forward gradients from clients and updates model parameters only when the variance falls below a predefined threshold.

FwdLLM adopts a centralized training paradigm where clients do not perform model updates, with all control logic and model updates conducted on the server. This centralized approach benefits from sufficient perturbations and alleviates non-IID issues, but exhibits a **critical limitation**: the server-side variance control mechanism severely constrains model update frequency. The server only updates when accumulated gradient variance falls below a predefined threshold, resulting in infrequent model updates that significantly slow convergence and require substantially more communication rounds to achieve target performance.

### 2.2.2 DeComFL: Client-Local Updates with Reconstruction Overhead

Different from FwdLLM, **DeComFL** focuses on reducing communication overhead in FL by utilizing forward training. Instead of transmitting full model gradients or parameters, DeComFL exchanges only the directional derivative $\nabla_v f(\theta)$ and the random seed of the perturbation vector $v$ between the server and clients. These elements allow all clients to reconstruct and aggregate gradients without explicit gradient sharing, achieving KB-level communication.

However, local updates in DeComFL require high-quality forward gradients, which demand extensive perturbations to ensure accurate gradient estimation. This requirement entails a **significant computational cost**. At each communication round, clients must revert and reconstruct model parameters based on updates from all previous rounds. As training progresses, especially for clients that have missed multiple rounds, the reconstruction process becomes increasingly computationally intensive due to the accumulation of historical updates. Additionally, DeComFL's local updates face challenges with non-IID data in FL, leading to unstable training dynamics where model accuracy may suddenly deteriorate and fail to converge.

## 2.3 An Overlooked Challenge: Many-Class Classification Tasks

Beyond the limitations of FwdLLM and DeComFL, another critical challenge has been entirely overlooked by existing forward training frameworks. Specifically, all prior works evaluate performance only on classification benchmarks with fewer than 10 classes, and we found that they perform poorly on tasks with many classes.

Our analysis reveals that existing forward training methods significantly lag behind backward training in head layer updates. In backward training, the head layer gradient norm is typically much larger

than other layers, especially in early training stages, and gradually decreases as the model converges. However, in forward training, gradients are determined by $v$ and $\nabla_v f(\theta)$, where $v$ is fixed-magnitude Gaussian noise and $\nabla_v f(\theta)$ is the projection of gradients onto direction $v$. When the head layer is untrained, classification predictions are nearly random, resulting in small differences in projections across different directions, which cannot provide accurate directional guidance, leading to slow convergence. This problem becomes more severe with more classes, requiring more perturbations to accumulate sufficiently accurate gradients. Our experiments with the FineFed-HP (Head Perturbation) variant in the appendix directly validate this analysis.

## 2.4 Summary of Requirements

Based on the analysis above, we identify the following key requirements for an effective forward-only FL framework:

- **Many-class classification capability:** Develop specialized mechanisms to overcome the performance degradation of existing forward methods on challenging many-class tasks, where inadequate head layer gradient estimation leads to slow convergence and poor accuracy.

- **Computational efficiency on edge devices:** Design perturbation strategies that minimize computational overhead while maintaining gradient quality, enabling practical deployment on resource-constrained clients.

- **Stable federated learning under non-IID data:** Enable client-side local updates while maintaining training stability under heterogeneous data distributions, avoiding the convergence issues that plague existing forward training methods in federated settings.

These requirements motivate the design of FineFed, which addresses each limitation through coordinated innovations in head tuning, uncertainty-guided perturbations, and shared momentum.

## 3 Methodology

Motivated by the limitations identified in the background, FineFed introduces three coordinated designs: (i) a forward-only head tuning mechanism that delivers accurate head gradients for many-class adaptation without autograd; (ii) an uncertainty-guided sample selection strategy with micro-batch perturbations that reduces client computation for PEFT parameters while preserving gradient quality; and (iii) a shared-momentum protocol that stabilizes frequent client-local updates under non-IID without server-side pacing.

## 3.1 Forward-Only Head Tuning for Many-Class Tasks

Many-class classification reveals a specific weakness of forward-only training under non-IID: the classification head must adapt rapidly to a large label space, yet generic forward-gradient estimators handle all parameters uniformly and often yield noisy, high-variance signals at the head. This often manifests as slow or unstable convergence on many-class vision tasks such as CIFAR-100 and TinyImageNet. Our first design therefore targets this bottleneck directly by delivering accurate head gradients without backward propagation or autograd.

**Manual computation of head gradients (FineFed-HT).** Consider a standard softmax–cross-entropy head with input $x \in \mathbb{R}^d$, parameters $(W, b)$, predicted probabilities $p = \mathrm{softmax}(Wx + b)$, and target $y$. Define $\Delta = p - \mathrm{onehot}(y)$. The loss gradients w.r.t. the head are

$$\nabla_b = \mathrm{mean}(\Delta), \qquad \nabla_W = \Delta^\top x, \tag{3}$$

which exactly match those obtained by back-propagation. These expressions can be computed using only forward operations if the head input $x$ is cached during inference. The implementation requires no autograd support and negligible additional memory (only $x$), making it practical on edge devices while providing accurate head signals for many-class adaptation; an implementation snippet is provided in Appendix B.

**Communication-efficient variant and trade-off.** For completeness, we also consider a head-perturbation variant (FineFed-HP) to attain KB-level communication under many-class settings; compared to FineFed-HT (our main method), it generally requires substantially more client computation (about 128–512 head perturbations on CIFAR-100/TinyImageNet) and attains lower accuracy. We keep FineFed-HT as the primary method and defer the detailed communication–accuracy–compute analysis to Sec. 4 and Appendix J.

## 3.2 UNCERTAINTY-GUIDED FORWARD GRADIENT ESTIMATION

Forward-gradient estimation becomes computationally onerous on edge devices when perturbed inference is applied uniformly across all samples. We address this by reducing the perturbation cost for estimating forward gradients for *PEFT parameters* $\theta_{\text{PEFT}}$ while maintaining gradient quality through two complementary mechanisms: (i) uncertainty-guided sample selection and (ii) micro-batch perturbations. The classification head is optimized via manual gradients in Sec. 3.1; in this section we focus on improving the efficiency of estimating forward gradients for PEFT parameters $\theta_{\text{PEFT}}$ with advanced workflow.

---

**Algorithm 1** FineFed: Uncertainty-Guided Forward Gradient Estimation for $\theta_{\text{PEFT}}$

---

**Require:** PEFT parameters $\theta_{\text{PEFT}}$, mini-batch of data $X$, number of selected samples $k$, number of micro-batches $M$, micro-batch size $b$, number of perturbation directions $P$
**Ensure:** Estimated forward gradient $g(\theta_{\text{PEFT}})$
    Compute losses $\mathcal{L}(x)$ for $x \in X$
    Select top-$k$ uncertain samples $X^*$
    Split $X^*$ into micro-batches $\{X_1, \ldots, X_M\}$ (where $M = \lceil |X^*|/b \rceil$), $|X_i| = b$
    Initialize $g(\theta_{\text{PEFT}}) \leftarrow 0$
    **for** each micro-batch $X_i$ **do**
        Reuse unperturbed loss $\mathcal{L}_0 = \frac{1}{b} \sum_{x \in X_i} \mathcal{L}(x)$
        **for** $p = 1$ to $P$ **do**
            Sample direction $v_p \sim \mathcal{N}(0, I)$
            Estimate forward gradient $g_p(\theta_{\text{PEFT}})$ with perturbed inference (Eq. 1 and Eq. 2)
            Accumulate $g(\theta_{\text{PEFT}}) \leftarrow g(\theta_{\text{PEFT}}) + g_p(\theta_{\text{PEFT}})$
        **end for**
    **end for**
**output** $g(\theta_{\text{PEFT}})$

---

**Uncertainty-guided sample selection.** Given a mini-batch, samples are ranked by cross-entropy loss and the top-$k$ uncertain samples are selected for perturbed inference. Concentrating the perturbation budget on informative data improves estimator efficiency and avoids redundant compute on confidently classified samples. Empirically, selecting approximately 25% of the samples attains accuracy comparable to using all samples while substantially reducing computation.

**Micro-batch perturbations.** The selected set is partitioned into micro-batches of size $b$ to align with edge-device throughput. For each micro-batch, the unperturbed loss is reused, directional derivatives across $P$ perturbation directions with respect to $\theta_{\text{PEFT}}$ are estimated, and the projected forward gradients are accumulated. The detailed algorithm is provided in Alg. 1.

Uncertainty-guided selection and micro-batching apply only to the PEFT parameters $\theta_{\text{PEFT}}$ to reduce perturbation cost. The classification head is updated with the manual gradients in Sec. 3.1 using *all* samples, which adds negligible cost and avoids selection bias at the head. Although sub-sampling could bias the forward-gradient for $\theta_{\text{PEFT}}$, we observed no statistically significant degradation under our settings.

## 3.3 LOCAL MODEL UPDATE WITH SHARED MOMENTUM

Forward-only federated fine-tuning faces a core tension between client-side and server-side updates. Client-side forward methods (e.g., FedZO, DeComFL) enable local updates but suffer from brittle, high-variance gradients—often requiring many perturbations and degrading under strong non-IID.

Server-side aggregation (e.g., FwdLLM) improves stability via centralized accumulation with cosine-similarity filtering and variance pacing, but it necessitates conservative step magnitudes and infrequent parameter updates, prolonging training and increasing synchronization overhead.

We adopt *shared momentum* as a system-level coupling mechanism for forward training in federated settings. Although methodologically standard (cf. FedAvg-M), it plays a central role in our forward-only framework: a globally shared momentum state aligns and regularizes client updates across rounds, reduces variance in local forward steps, and enables stable local optimization without centralizing gradients. From a framework viewpoint, shared momentum is simple to integrate and provides a persistent control signal that makes forward-only local updates practical and reliable.

---

**Algorithm 2** FineFed: Local Update with Shared Momentum

---

1: Initialize $m^0 = 0$
2: **for** each round $r = 0..R - 1$ **do**
3:     Server selects $N$ clients; broadcast $(\theta^r, m^r)$
4:     **for** each client $i$ in parallel **do**
5:         Initialize $(\theta_i^{r,0}, m_i^{r,0}) = (\theta^r, m^r)$
6:         **for** $t = 1..T$ **do**
7:             Compute gradient $g(\theta_i^{r,t})$ with head tuning (Eq. 3) and uncertainty-guided forward gradient estimation (Alg. 1)
8:             $m_i^{r,t} = \beta\, m_i^{r,t-1} + g(\theta_i^{r,t})$
9:             $\theta_i^{r,t} = \theta_i^{r,t-1} - \eta\, m_i^{r,t}$
10:         **end for**
11:         Upload $(\theta_i^{r,T}, m_i^{r,T})$
12:     **end for**
13:     Server aggregates parameters and momentum via FedAvg
14: **end for**

---

Operationally, in round $r$ the server broadcasts $(\theta^r, m^r)$. Each selected client performs $T$ local steps where it (i) computes *manual head gradients* via Eq. 3 and (ii) computes *uncertainty-guided forward gradients* for $\theta_{\mathrm{PEFT}}$ via Sec. 3.2 (Alg. 1); these are combined into $g$ to update momentum and parameters $(m, \theta)$ as in Alg. 2. Clients return $(\theta_i^{r,T}, m_i^{r,T})$, and the server averages both to obtain $(\theta^{r+1}, m^{r+1})$. Sharing $m^r$ aligns client directions with previously effective updates, improving stability and convergence under heterogeneity.

Shared momentum is orthogonal to forward-only head tuning (Sec. 3.1) and the uncertainty-guided estimator for $\theta_{\mathrm{PEFT}}$ (Sec. 3.2). With compact PEFT states, communicating momentum adds modest bytes yet yields substantial stability gains; we use $\beta{=}0.9$ with standard FL schedules. Empirically, the shared momentum of FineFed outperforms FwdLLM's cosine-similarity filtering and variance pacing while remaining simpler and more scalable. We provide additional comparisons in Appendix H.

**Summary.** Together, these mechanisms yield a cohesive forward-training framework for federated learning that enables exact and scalable head optimization for many-class tasks, reduces client-side computation for forward-gradient estimation, and strengthens convergence under non-IID heterogeneity, constituting a more effective system design than existing forward/ZO-in-FL approaches.

## 4 EVALUATION

We organize this section around the following research questions, which structure our empirical study:

- **RQ1 (Accuracy and Convergence).** How does *FineFed* compare to forward- and backward-training baselines in accuracy and convergence across NLP and vision tasks?

- **RQ2 (System Efficiency).** What are the per-round communication, client-side computation, and memory trade-offs of *FineFed* relative to alternative methods?

- **RQ3 (Effectiveness of Key Designs).** What are the contributions of forward-only head tuning, uncertainty-guided sampling, and shared momentum to overall performance?

## 4.1 EXPERIMENTAL SETUP

**Models and Datasets.** We employ RoBERTa (Liu et al., 2019) and ViT-B/16 (Wu et al., 2020) for NLP and vision tasks, respectively. We evaluate on two NLP benchmarks (SST-2, AGNEWS) and three vision benchmarks (CIFAR-10/100, TinyImageNet), with additional SuperGLUE results in Appendix D. We consider three PEFT methods (*Adapter*, *LoRA*, *BitFit*) with approximately 1 MB of trainable parameters ( 0.3% of full fine-tuning).

**Federated Settings.** We simulate 100 clients under a non-IID label-skew using the Pathological Partition strategy (Li et al., 2024a; Oh et al., 2022), with $s$ classes per client ($s$=5 for CIFAR-10; $s$=50 for CIFAR-100 and TinyImageNet). Training runs for 100 rounds with 5 clients per round and 1 local epoch per client. We also study an extreme heterogeneity setting with $s$=5 on CIFAR-100.

**Baselines.** We compare against FedAvg (McMahan et al., 2017), FedAvg-M (Cheng et al., 2024), DeComFL (Li et al., 2024b), and FwdLLM (Xu et al., 2024). Detailed configurations are provided in Appendix C.

## 4.2 RQ1: ACCURACY AND CONVERGENCE

We first assess accuracy and convergence across tasks and PEFT variants, benchmarking *FineFed* against forward- and backward-training baselines.

Table 1: Best test accuracy (%) achieved by FineFed and baselines for fine-tuning RoBERTa-base model with BitFit and ViT-B/16 model with various PEFT methods. Number with * means the best of all baselines, underline is second, and **bold** is the best of forward training methods.

| Dataset | SST-2 | AGNEWS | CIFAR-10 | | | CIFAR-100 | | | TinyImageNet | | |
|---|---|---|---|---|---|---|---|---|---|---|---|
| PEFT | | BitFit | Adapter | LoRA | BitFit | Adapter | LoRA | BitFit | Adapter | LoRA | BitFit |
| FedAvg | 90.2 | 89.7 | 98.1 | 97.6 | 98.1 | 86.8 | 87.2 | 88.2 | 83.2 | 83.6 | 84 |
| FedAvg-M | 91.1* | 92.1* | 98.5* | 98.4 | 98.3 | 91.3* | 90.4 | 90.2 | 87.4* | 85.7 | 87.2 |
| FwdLLM | 87.6 | 86.7 | 92.3 | 92.2 | 95.3 | 22.3 | 19.7 | 20 | 6.2 | 7.6 | 7.1 |
| DeComFL | 87.4 | 86.9 | 85.7 | 87.6 | 94.5 | 20.2 | 19.4 | 23.4 | 6.1 | 8.7 | 7.2 |
| FineFed | **89.0** | **88.4** | 96.9 | 96.3 | **97.5** | 86.1 | 84.2 | **88.0** | 84.1 | 82.3 | **84.9** |

**Model Accuracy.** Table 1 demonstrates *FineFed*'s competitive performance across all datasets. Notably, *FineFed* outperforms all forward training baselines and even surpasses the traditional backward-based FedAvg on TinyImageNet (84.9% vs 84.0%). Across PEFT choices, BitFit is the best-performing method; this pattern is consistent across different forward-training algorithms and aligns with the observations reported in FwdLLM. We hypothesize that BitFit's much smaller number of trainable parameters makes it particularly well-suited to zeroth-order, perturbed-inference gradient estimation.

**Extreme Non-IID Evaluation.** We also evaluate under an extreme non-IID setting (5 classes per client on CIFAR-100): *FineFed* reaches 87.6% test accuracy within 80 rounds, whereas FwdLLM and DeComFL are below 5%. See Appendix I for details.

**Hybrid Training Enhancement.** *FineFed* can be enhanced through hybrid training with backward-capable clients (e.g., laptops or high-end mobile devices): *FineFed-Hybrid* with one backward-capable client achieved 85.8% test accuracy on TinyImageNet with BitFit, which is +0.9% better than *FineFed* with forward-only clients (84.9%), and performance gradually approaches FedAvg-M as the proportion of backward-capable clients increases.

**Integration with Advanced Zeroth-Order Optimizers.** We also explore integrating advanced zeroth-order optimization methods with *FineFed*. Specifically, we develop *FineFed-HiZOO* by combining HiZOO (Zhao et al., 2025) with *FineFed*. While *FineFed-HiZOO* achieves competitive performance (97.0/85.5/81.3 on CIFAR-10/100/TinyImageNet), it does not outperform the standard *FineFed*; design details and analysis are deferred to Appendix L.

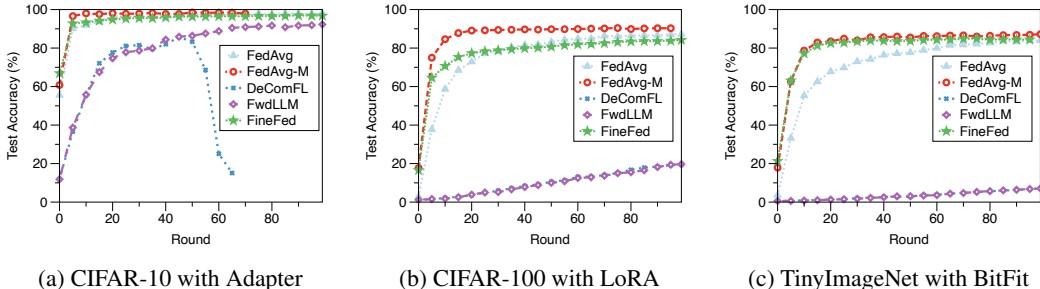

Figure 1: Test accuracy for fine-tuning ViT-B/16 model with PEFT.

**Convergence Analysis.** Figure 1 reveals a key limitation of existing forward training methods: while FwdLLM and DeComFL perform reasonably on simple tasks (CIFAR-10), they fail catastrophically on complex many-class scenarios (CIFAR-100, TinyImageNet). In contrast, *FineFed* maintains robust convergence across all task complexities, achieving speeds comparable to backward training methods. Appendix G provides round-to-target convergence results together with a detailed analysis.

## 4.3 RQ2: SYSTEM EFFICIENCY

We next quantify system-level efficiency along three axes: per-round communication, client-side computation, and memory footprint.

**Per-round Communication Overhead.** We quantify the per-round bytes sent (uplink) and received (downlink) between clients and server. Table 2 summarizes per-round costs. Parameter-exchange methods (FedAvg, FedAvg-M, FineFed) operate at MB-scale because they transmit trainable parameters (and momentum) each round. Pure ZO-based schemes (FwdLLM, DeComFL) communicate at KB-scale by sending seeds/projections, with downlink scaling in $N$ (number of selected clients). FineFed-HP replaces manual head-gradient computation with head perturbations. This makes it a pure ZO variant and enables KB-level communication via a DeComFL-like reconstruction mechanism (send seeds/projections; reconstruct on devices). However, the cost is nearly doubled computation and a noticeable accuracy reduction (e.g., 1.5–2.0 percentage points on CIFAR-100/TinyImageNet); a comprehensive analysis is provided in Appendix J.

Table 2: Per-Round Communication Costs Comparison (ViT-B/16 with BitFit, $\|\theta\| \approx 1$ MB)

| Method | Uplink | Downlink | Total | Notes |
|---|---|---|---|---|
| FedAvg | $\|\theta\|$ | $\|\theta\|$ | 2 MB | Parameters only |
| FedAvg-M | $\|\theta\|$ | $2\|\theta\|$ | 3 MB | Parameters + momentum |
| FwdLLM | $2P$ | $\|\theta\|$ | 1 MB | $P$ seeds + $P$ projections |
| DeComFL | $1 + P$ | $N(1 + P)$ | 2 KB | 1 seed + $P$ projections |
| **FineFed** | $\|\theta\|$ | $2\|\theta\|$ | 3 MB | Same as FedAvg-M |
| **FineFed-HP** | $1 + kP$ | $N(1 + kP)$ | 2 KB | Head perturbation variant |

**Computation Cost of Gradient Estimation.** We present a quantitative analysis of the per-client, per-round computation cost measured in sample-level units, where one unit equals the FLOPs of a single forward pass. Let $B$ denote the batch size and $P$ the number of perturbations per sample. For FedAvg, the cost comprises $B$ forward passes plus $\approx 2B$ for the backward pass (total $\approx 3B$). For FwdLLM, each perturbation is evaluated in both the positive and negative directions (symmetric difference), resulting in a total of $2BP$ units. For DeComFL and FineFed (one-sided difference), each sample is first evaluated without perturbation and then along $P$ perturbation directions per sample. In addition, FineFed applies perturbations only to the top-$k$ most uncertain samples, yielding $B + kP$ units. The best computation cost for each method is summarized in Table 3.

**Client-side Memory Footprint.** The memory overhead introduced by forward training methods is minimal compared to normal inference. *FineFed* has a memory footprint comparable to other forward training methods, requiring only a few MB to store the shared momentum and cache the input of the

Table 3: Per-client, per-round computation cost ($B=64$). See Appendix F for a budget sweep.

| Method | Inference Cost | Perturbation Cost | Best Configuration | Total Cost |
|---|---|---|---|---|
| FedAvg | $B$ | $\approx 2B$ | — | $\approx 192$ |
| FwdLLM | — | $B \times 2 \times P$ | $P=5$ | 640 |
| DeComFL | $B$ | $B \times P$ | $P=10$ | 704 |
| FineFed | $B$ | $k \times P$ | $P=8, k=16$ | 192 |

head layer. This is significantly more efficient than backward training methods, which require storing all activations for back-propagation. Detailed memory footprint analysis is provided in Appendix E.

### 4.4 RQ3: EFFECTIVENESS OF KEY DESIGNS

Finally, we perform ablations to isolate the contributions of each design—forward-only head tuning, uncertainty-guided sampling, and shared momentum—on convergence and accuracy.

Table 4: Ablation Study: best test accuracy (%) achieved on various datasets using ViT-B/16 with BitFit.

| Dataset | CIFAR-10 | CIFAR-100 | TinyImageNet |
|---|---|---|---|
| FineFed | 97.5 | 88.0 | 84.9 |
| w/o Momentum | 95.8 | 80.4 | 80.8 |
| w/ Random Sampling | 97.0 | 87.4 | 83.0 |
| w/ All Samples[*] | 97.8 | 88.5 | 85.0 |

[*] training with all samples requires $4\times$ of computation

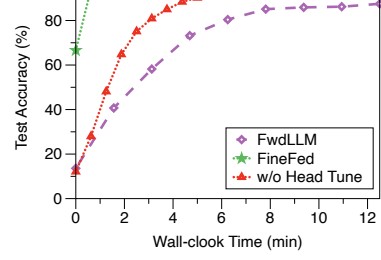

Figure 2: Ablation study of Head Tuning on CIFAR-10 with BitFit.

**Forward-only head tuning** is essential for many-class adaptation. Figure 2 shows that removing head tuning markedly slows FineFed's convergence; nevertheless, in wall-clock time, FineFed (w/o Head Tune) remains about $2.5\times$ faster than FwdLLM. This aligns with Section 3: forward-only head tuning supplies exact, low-cost gradients for the classification head without autograd, thereby overcoming optimization difficulties in the many-class regime.

**Uncertainty-guided sampling** strikes a strong efficiency–accuracy balance. Replacing it with random sampling (w/ Random Sampling) degrades accuracy by up to 1.9 percentage points on TinyImageNet; using all samples (w/ All Samples) yields only modest gains (e.g., CIFAR-100 +0.5) at roughly $4\times$ computation. This indicates that targeted selection under a fixed compute budget is markedly more cost-effective than naively increasing perturbations.

**Shared momentum** yields substantial gains and is particularly critical under forward training. As shown in Table 4, relative to removing momentum (w/o Momentum), FineFed improves by +1.7/+7.6/+4.1 percentage points on CIFAR-10/CIFAR-100/TinyImageNet. Beyond its known benefits for FedAvg, shared momentum couples client directions across rounds, damping the high-variance noise from zeroth-order estimation and non-IID skew, thereby stabilizing and accelerating convergence. Further momentum-focused ablations are provided in Appendix H.

### 5 CONCLUSION

We introduced *FineFed*, a forward-only federated fine-tuning framework for resource-constrained clients. By coupling forward-only head tuning with uncertainty-guided perturbed inference and shared momentum, FineFed makes forward training practical in federated learning. Across NLP and vision tasks, it surpasses prior forward methods and approaches the accuracy and convergence of backward training, including on many-class tasks and under extreme non-IID settings, while preserving the low-memory footprint and reducing client computation. This points to a scalable path for deploying large-model FL in heterogeneous, real-world environments.

## 6 ETHICS STATEMENT

We acknowledge our adherence to the ICLR Code of Ethics. This work presents *FineFed*, a federated learning framework that enables efficient fine-tuning of large models across resource-constrained devices. Our research addresses privacy and security considerations inherent in federated learning by reducing communication overhead and supporting parameter-efficient fine-tuning methods.

The proposed method does not involve human subjects or sensitive personal data collection. All experiments are conducted on standard benchmark datasets (CIFAR-10, CIFAR-100, TinyImageNet, SuperGLUE) commonly used in computer vision and NLP research. We have not identified any potential for harmful applications or misuse beyond standard federated learning scenarios.

Our work contributes to the democratization of AI by enabling participation of resource-constrained devices in federated learning, potentially reducing computational barriers and environmental impact. We recognize the importance of continued research into security, fairness, and robustness in federated systems, which remain active areas of investigation in the field.

## 7 REPRODUCIBILITY STATEMENT

To ensure reproducibility of our work, we provide comprehensive implementation details and experimental configurations throughout the paper. Our complete source code, including the FineFed framework implementation and all experimental scripts, is available at `https://anonymous.4open.science/r/FineFed-0554/`. The implementation is built on PyTorch and follows the NIID-Bench framework standards for federated learning experiments. Detailed experimental settings, including model architectures (RoBERTa-base, ViT-B/16), datasets (SST-2, AGNEWS, CIFAR-10/100, TinyImageNet, SuperGLUE), PEFT configurations (Adapter, LoRA, BitFit), and hyperparameters are provided in Appendix C. All training hyperparameters, including learning rates, momentum settings, and forward training parameters ($T$, $P$, $h$), are specified in Table 7. The manual head-gradient computation code snippet is provided in Appendix B, demonstrating the forward-only implementation. Additional experimental results, including SuperGLUE benchmarks, memory footprint analysis, convergence studies, and ablation analyses, are detailed in the appendix. The code repository includes scripts for reproducing all experiments, data preprocessing pipelines, and evaluation metrics used in our study.

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

## A    NOTATION AND SYMBOLS

Table 5: Symbols and their meanings used throughout the paper.

| Symbol | Meaning |
|---|---|
| $\theta$ | Model parameters (trainable subset in PEFT settings) |
| $\theta_{\text{PEFT}}$ | Trainable PEFT parameters (e.g., Adapter/LoRA/BitFit) |
| $(W, b)$ | Classification head weight and bias |
| $x$ | Cached input (feature) to the classification head |
| $p = \text{softmax}(Wx + b)$ | Predicted class probabilities |
| $y$ | Ground-truth label |
| $\mathcal{L}$ | Loss (cross-entropy unless noted) |
| $v \sim \mathcal{N}(0, I)$ | Random perturbation direction for forward gradients |
| $g_v(\theta)$ | Forward (zeroth-order) gradient estimator (Eq. 2) |
| $h$ | Smoothing factor/step for finite differences |
| $B$ | Mini-batch size |
| $k$ | Number of selected uncertain samples per mini-batch |
| $M$ | Number of micro-batches formed from the $k$ samples |
| $b$ | Micro-batch size |
| $P$ | Number of perturbation directions per (micro-)batch |
| $P_{\text{head}}$ | Number of perturbation directions per head in FineFed-HP |
| *budget* | Compute budget for perturbations, defined as $k \times P$ |
| $T$ | Number of local updates per round |
| $\eta$ | Learning rate |
| $m$ | Momentum state (shared between server and clients) |
| $N$ | Number of selected clients per communication round |
| $s$ | Number of classes per client under label-skew partitioning |
| $\|\theta\|$ | Size of transmitted trainable parameters (bytes/MB) |

## B    IMPLEMENTATION DETAILS

We implement the framework of *FineFed* and a federated learning framework using PyTorch. We implement the training process of *FineFed* with forward-only computation – no `backward` or `autograd` are used in FineFed. The federated learning framework is implemented based on NIID-Bench (Li et al., 2021) to ensure our framework is aligned with the literature standards of federated learning.

**Manual head-gradient snippet (from Sec. 3.1).**   The following PyTorch-style snippet implements the exact softmax–cross-entropy head gradients described in Sec. 3.1 using only forward operations:

```
logits = w_head @ input_head + b_head
p = F.softmax(logits)
diff = p - F.one_hot(labels, num_class)
grad = diff / batch_size
b_grad = grad.sum(dim=0)
w_grad = grad.T @ input_head
```

## C    EXPERIMENTAL CONFIGURATION

### C.1    MODELS AND DATASETS

**Base Models.**   For NLP tasks, we employ RoBERTa-Base (Liu et al., 2019) pre-trained on 58M tweets and finetuned for emotion recognition with the TweetEval benchmark (Barbieri et al., 2020).

For vision tasks, we use ViT-B/16 (Wu et al., 2020) pre-trained on ImageNet-21k by Google Research (Dosovitskiy et al., 2021). ViT introduced a pioneering transformer-based approach to computer vision by dividing input images into fixed-size patches, linearly embedding each patch, and sequentially processing these embeddings through standard transformer encoder layers. This architecture enables the simultaneous capture of both local spatial features and long-range global dependencies within visual data.

**Datasets.** We evaluate on six datasets across NLP and vision domains:

- **SST-2** (Socher et al., 2013): Binary sentiment classification with 67,000 movie review sentences
- **AGNEWS** (Zhang et al., 2015): News topic classification with 120,000+ training samples across 4 topics (World, Sports, Business, Sci/Tech)
- **SuperGLUE** (Wang et al., 2019): Multi-task NLP benchmark with 5 diverse tasks (CB, RTE, WIC, WSC, BoolQ) - results reported in Appendix D
- **CIFAR-10/100** (Krizhevsky et al., 2009): 60,000 $32\times32$ RGB images with 10/100 classes (6,000/600 images per class)
- **TinyImageNet** (Le & Yang, 2015): 200 classes of $64\times64$ images (500 training, 50 validation, 50 test per class)

For federated training, all datasets are partitioned into training and test sets, distributed across clients to emulate realistic FL scenarios. We simulate a federated system with 100 clients under a non-IID, label-skewed distribution using the Pathological Partition strategy (Li et al., 2024a; Oh et al., 2022). Each client is assigned $s$ distinct classes ($s = 5$ for CIFAR-10 and $s = 50$ for CIFAR-100 and TinyImageNet), ensuring balanced data volumes. Training proceeds for 100 communication rounds, with five clients selected to train for one local epoch per round. All experiments are conducted on an NVIDIA A30 GPU to simulate a distributed federated environment.

### C.2 PARAMETER-EFFICIENT FINE-TUNING (PEFT) METHODS

To assess the performance of *FineFed* under various PEFT strategies, we evaluate three widely adopted methods for fine-tuning Vision Transformers (ViTs). PEFT methods enable efficient adaptation of large pre-trained models by updating only a small subset of parameters, significantly reducing computational and memory requirements while maintaining competitive performance.

- **Adapter**: Modules inserted at the output of each transformer layer, with a hidden dimension size of 8. These modules consist of a down-projection layer, a non-linear activation function, and an up-projection layer, providing task-specific adaptation while maintaining the original model's capacity. The adapter modules are applied after the multi-head attention and feed-forward layers, allowing for flexible adaptation without modifying the core transformer architecture.
- **LoRA** (Low-Rank Adaptation): Applied to the query and value projection matrices in the self-attention layers, with a rank size of 8. This approach enables efficient adaptation by learning low-rank updates to the attention weights. LoRA decomposes the weight update as $\Delta W = BA$, where $B \in \mathbb{R}^{d\times r}$ and $A \in \mathbb{R}^{r\times k}$ with rank $r \ll \min(d, k)$. This factorization dramatically reduces the number of trainable parameters while preserving the model's representational capacity.
- **BitFit**: Fine-tunes only the bias terms across all layers. This method provides the most parameter-efficient adaptation by updating only the bias parameters while keeping all other weights frozen. Despite its simplicity, BitFit has shown remarkable effectiveness, particularly in scenarios with limited computational resources, as bias parameters often contain task-specific information that can be efficiently adapted.

The parameter breakdown for FineFed with different PEFT methods is detailed in Table 6. FineFed always combines head tuning (0.6 MB) with a PEFT method, as head tuning alone would be equivalent to traditional FedAvg and cannot achieve the performance improvements needed for complex tasks. The additional parameters introduced by each PEFT method range from 0.4 MB (BitFit) to 1.1

MB (LoRA), with total trainable parameters ranging from 1.0 MB to 1.7 MB, representing only 0.2-0.3% of full fine-tuning parameters. This dramatic reduction in trainable parameters makes FineFed particularly suitable for federated learning scenarios, where communication efficiency and client-side resource constraints are critical considerations.

Table 6: Trainable parameter breakdown for FineFed with different PEFT methods on ViT-B/16. Head tuning (0.6 MB) is the base component, with PEFT methods adding additional parameters for effective adaptation.

| Component | Head Tune | Adapter | LoRA | BitFit |
|---|---|---|---|---|
| Base (Head Tune) | 0.6 MB | 0.6 MB | 0.6 MB | 0.6 MB |
| PEFT Addition | – | +0.6 MB | +1.1 MB | +0.4 MB |
| **Total** | **0.6 MB** | **1.2 MB** | **1.7 MB** | **1.0 MB** |
| % of Full FT | 0.1% | 0.2% | 0.3% | 0.2% |

**Parameter Efficiency Impact.** The dramatic reduction in trainable parameters directly influences both the memory footprint and the communication overhead in federated settings such as *FineFed*. As shown in Table 6, FineFed configurations substantially reduce the number of parameters requiring updates, with total trainable parameters ranging from 1.0 MB (head + BitFit) to 1.7 MB (head + LoRA), representing only 0.2-0.3% of the full fine-tuning size. This parameter efficiency has several important implications:

- **Memory Efficiency**: PEFT methods require significantly less GPU memory during training, as only a small subset of parameters need to be stored and updated. This is particularly crucial for edge devices with limited memory capacity.

- **Communication Efficiency**: In federated learning, the reduced parameter count translates directly to lower communication costs, as clients only need to transmit and receive the small PEFT parameter updates rather than the entire model.

- **Computational Efficiency**: Training with fewer parameters reduces the computational overhead, making it feasible to perform local updates on resource-constrained devices.

- **Storage Efficiency**: PEFT methods enable efficient model storage and deployment, as the base model remains unchanged while only the small adapter components need to be stored and loaded.

The choice of PEFT method involves a trade-off between parameter efficiency and adaptation capacity. BitFit offers the highest parameter efficiency (0.1%) but may have limited adaptation capacity for complex tasks. LoRA provides a good balance between efficiency and capacity, while Adapter offers more flexibility at the cost of slightly more parameters.

In our experiments, we find that all three methods perform effectively with *FineFed*, with BitFit often achieving the best performance.

## C.3 TRAINING HYPERPARAMETERS

**Optimizer Settings.** We adopt the SGD optimizer with a fixed learning rate of 0.1 across all baselines and PEFT methods. For FedAvg-m, we follow their standard setup and use a momentum of 0.5 with a damping factor of 0.5. For *FineFed*, we use a momentum of 0.9 and set the damping factor to 0, which empirically improves convergence.

**Training Configuration.** For backward training methods (FedAvg, FedAvg-M), clients use SGD with a mini-batch size of 64, learning rate 0.1, and momentum 0.9. For forward training methods, clients perform perturbed inferences with a batch size of 8, uploading forward gradients to the server. The server then aggregates and applies these gradients using a learning rate of 0.01 for FwdLLM and DeComFL, adapted from their original implementations for stability.

**Forward Training Parameters.** Forward training is governed by three key parameters: the number of local updates $T$ (i.e., mini-batches), the number of perturbations $P$, and the smooth factor $h$. Both

*FineFed* and DeComFL perform $T$ local model updates, where gradients are computed by averaging over $P$ forward perturbations. *FineFed* further enhances gradient estimation by selecting the top-$k$ most uncertain samples at the micro-batch level, as detailed in Algorithm 1.

In contrast, FwdLLM does not execute local updates on clients. Instead, it aggregates forward gradients on the server, accumulating a total of $TP$ perturbations per client. We set both $T$ and $P$ to 5 for FwdLLM, 25 perturbations per client. However, the total number of perturbations is dynamically determined by a variance-based pacing strategy that halts accumulation once gradient variance falls below a threshold. We set the variance threshold to 0.2 according to their open-source implementation[1].

For the smooth factor $h$, we set it to 0.001 for all forward training methods. The detailed hyperparameter settings used in our experiments are summarized in Table 7.

Table 7: Summary of Experimental Settings

| Method | SGD Optimizer | | | Forward Training | | | |
|--------|---------------|----------|---------|---|----|------|------------------|
| | Learning Rate | Momentum | Damping | $T$ | $P$ | $h$ | Special Settings |
| FedAvg | 0.1 | – | – | 5 | – | – | – |
| FedAvg-m | 0.1 | 0.5 | 0.5 | 5 | – | – | – |
| DeComFL | 0.1 | – | – | 5 | 10 | 1e-3 | – |
| FwdLLM | 0.1 | – | – | 5 | 5 | 1e-3 | variance threshold = 0.2 |
| FineFed | 0.1 | 0.9 | 0.0 | 5 | 8 | 1e-3 | micro-batch size = 8, top-$k$ = 16 |

# D  MORE RESULTS ON SUPERGLUE

We further evaluate *FineFed* and other baselines on the SuperGLUE benchmark, which consists of five diverse NLP tasks: **CB** (CommitmentBank) for hypothesis inference, **WSC** (Winograd Schema Challenge) for commonsense reasoning, **WIC** (Word-in-Context) for word sense disambiguation, **RTE** (Recognizing Textual Entailment), and **BoolQ** for question answering.

Datasets and prompts are prepared following the official DeComFL implementation[2]. We set the maximum sequence length to 512 for BoolQ and 128 for the other datasets.

In addition to the settings described in Appendix C, we apply the following changes for SuperGLUE:

- Reduce the total number of clients to 10, with 5 selected per communication round.

- For the CB dataset, due to its small training size (only 250 samples), we use 5 clients in total and sample 2 per round.

- Learning rates are set to $1 \times 10^{-2}$ for BitFit and $1 \times 10^{-4}$ for LoRA.

Table 8: Test accuracy (%) of PEFT methods and baselines on SuperGLUE datasets. Each column represents performance on one dataset, under either BitFit or LoRA adaptation for PEFT methods.

| Dataset | CB | | RTE | | WIC | | WSC | BoolQ |
|---------|--------|------|--------|------|--------|------|--------|--------|
| PEFT | BitFit | LoRA | BitFit | LoRA | BitFit | LoRA | BitFit | BitFit |
| FedAvg | 81.5 | 81.6 | 58.7 | 66.2 | 57.1 | 64.2 | 62.9 | 65.8 |
| FedAvg-m | 81.4 | 83.7 | 58.7 | 67.3 | 58.4 | 64.6 | 63.9 | 65.2 |
| FwdLLM | 76.2 | 57.1 | 56.5 | 51.6 | 54.5 | 54.9 | 63.9 | 62.2 |
| DeComFL | 60.7 | 57.1 | 48.3 | 51.3 | 46.7 | 52.2 | 59.8 | 62.1 |
| FineFed | **83.7** | 71.6 | 55.7 | 52.0 | **58.6** | 57.1 | 61.9 | 65.4 |

---

[1] https://github.com/UbiquitousLearning/FwdLLM
[2] https://github.com/ZidongLiu/DeComFL/blob/main/cezo_fl/util/language_utils.py

The test accuracy results (%) on each dataset are summarized in Table 8. Notably, ***FineFed* achieves the highest accuracy on CB and WIC**, demonstrating its effectiveness in tasks involving hypothesis inference and word sense disambiguation. *FineFed* is highly competitive across other tasks as well, achieving performance comparable to or better than existing baselines, including FedAvg and FedAvg-m. These results underscore the strength of *FineFed* in balancing computation and communication efficiency with model performance in federated learning scenarios.

## E  MEMORY FOOTPRINT ANALYSIS

Table 9 provides a detailed comparison of memory footprint for fine-tuning ViT-B/16 with different PEFT methods. The results show that forward training methods (FwdLLM, DeComFL, and FineFed) have significantly lower memory requirements compared to traditional backward training methods like FedAvg.

Table 9: Comparison of Memory Footprint for Fine-Tuning ViT-B/16 with Different PEFT Methods

| Method | Adapter | LoRA | BitFit |
|---|---|---|---|
| FedAvg | 1136.1 MB | 1109.3 MB | 1066.5 MB |
| FwdLLM/DeComFL | 404.6 MB | 405.6 MB | 403.8 MB |
| FineFed | 405.8 MB | 407.3 MB | 404.8 MB |

The memory overhead for forward training methods is minimal compared to normal inference. FwdLLM and DeComFL store the accumulated gradients with memory overhead of $\|\theta\|$. *FineFed* requires additional memory for storing the shared momentum, resulting in memory overhead of $2 \times \|\theta\|$. In contrast, backward training requires storing all activations for back-propagation, which is about $3 \times \|\theta\|$ for fine-tuning ViT-B/16 with PEFT.

## F  COMPUTATION BUDGET AND MICRO-BATCH PERTURBATION

This section extends the fixed-configuration compute analysis in the main text by sweeping the perturbation *budget* across settings for CIFAR-10/100 (ViT-B/16 + BitFit). We follow the same notation as Section 4: batch size $B$, perturbations per sample $P$, and selected samples $k$ (top-$k$). The budget is

$$\text{budget} = k \times P,$$

where $k$ is the number of selected samples per round. In our micro-batch implementation, $k = M \cdot b$.

We report the rounds to reach the target accuracy and the best accuracy obtained under each budget. This complements the main text by showing how efficiency and accuracy trade off beyond the single configuration reported there.

Table 10: CIFAR-10: budget vs. convergence and best accuracy. Target: 97% test accuracy.

| budget | # micro-batch | mb size | # perturb | # rounds (>97%) | best acc (%) |
|---|---|---|---|---|---|
| 64 | 1 | 8 | 8 | 45 | 97.4 |
| 128 | 2 | 8 | 8 | 35 | 97.5 |
| 256 | 2 | 8 | 16 | 30 | 97.7 |
| 512 | 2 | 16 | 16 | 20 | – |

Overall, a budget of 128 (e.g., $M{=}2$, $b{=}8$, $P{=}8$) yields strong accuracy with good convergence speed for *FineFed*, while increasing the budget to 256 brings marginal gains at doubled perturbation compute.

## G  ROUND-TO-TARGET CONVERGENCE ANALYSIS

We report round-to-target convergence results across datasets and task complexities, and provide accompanying analysis comparing *FineFed* with baseline methods.

Table 11: CIFAR-100: budget vs. convergence and best accuracy. Target: 87% test accuracy.

| budget | # rounds (>87%) | best acc (%) |
|--------|-----------------|--------------|
| 64 | 75 | 87.4 |
| 128 | 65 | 88.0 |
| 256 | 55 | 88.2 |
| 512 | 40 | 88.8 |

Table 12 summarizes the number of communication rounds required to reach target accuracy thresholds on each dataset.

Table 12: Convergence comparison: Number of communication rounds required to reach target accuracy thresholds.

| Method | CIFAR-10 (>97%) | CIFAR-100 (>85%) | TinyImageNet (>84%) |
|--------|-----------------|-------------------|----------------------|
| FedAvg | 20 | 90 | 75 |
| FedAvg-M | 15* | 35* | 45* |
| FwdLLM | – | – | – |
| DeComFL | – | – | – |
| **FineFed** | **35** | **40** | **55** |

These results yield several observations:

- **Forward training failure**: FwdLLM and DeComFL fail to converge on complex many-class tasks (CIFAR-100, TinyImageNet), indicated by "–" in the table

- **FineFed's efficiency**: While FineFed may require more rounds than FedAvg-M, it converges on all tasks where forward-training baselines fail

- **Task complexity scaling**: FineFed's required rounds scale reasonably with task complexity ($35 \rightarrow 40 \rightarrow 55$)

# H  ABLATION ON SHARED MOMENTUM

To understand the contribution of each FineFed component, we conduct an ablation study comparing FineFed with shared momentum only versus the full system. Table 13 shows the convergence behavior on CIFAR-10.

Table 13: Ablation on shared momentum: Convergence comparison on CIFAR-10 (ViT-B/16 + BitFit) for FwdLLM, FineFed with momentum only, and the full FineFed system.

| Round | FwdLLM | FineFed (Momentum Only) | FineFed (Full) | Improvement |
|-------|--------|--------------------------|----------------|-------------|
| 0 | 14.9 | 12.3 | 66.6 | +4.5× |
| 5 | 52.5 | 34.9 | 94.3 | +1.8× |
| 10 | 59.8 | 54.7 | 95.4 | +1.7× |
| 20 | 85.8 | 78.5 | 96.5 | +1.1× |
| 40 | 90.4 | 89.6 | 97.1 | +1.1× |
| 80 | 92.2 | 94.4 | 97.4 | +1.1× |

This analysis highlights that shared momentum by itself confers stability and steady gains over FwdLLM, while the full *FineFed* system yields pronounced early acceleration and sustains a performance advantage across all rounds.

## I    EXTREME NON-IID PERFORMANCE ANALYSIS

To evaluate the robustness of *FineFed* under extreme data heterogeneity, we conduct experiments on CIFAR-100 with only 5 distinct classes per client ($s = 5$), creating a highly challenging non-IID scenario. This extreme setting tests the ability of forward training methods to handle severe data distribution skew, which is common in real-world federated learning deployments.

### I.1    EXPERIMENTAL SETUP

We maintain the same experimental configuration as described in Appendix C, with the following modifications for extreme non-IID evaluation:

- **Dataset**: CIFAR-100 with only 5 classes per client (reduced from 50)
- **Model**: ViT-B/16 with BitFit PEFT method
- **Training rounds**: 100 communication rounds
- **Client selection**: 5 clients per round
- **Local epochs**: 1 epoch per client

### I.2    RESULTS AND ANALYSIS

Table 14 summarizes performance under extreme non-IID conditions.

**Forward Training Collapse.** Both FwdLLM and DeComFL completely fail to converge in this setting, achieving less than 5% accuracy. This catastrophic failure highlights fundamental limitations of existing forward training methods under severe data heterogeneity.

**FineFed's Robustness.** In contrast, *FineFed* attains 87.6% accuracy—substantially outperforming forward-only baselines and approaching backward-based methods (e.g., FedAvg-M at 89.1%) despite using forward-only training. Consistent with the ablation in Appendix H (momentum-only vs. full), these gains under $s=5$ are driven primarily by the shared momentum mechanism.

Table 14: Best test accuracy (%) achieved under extreme non-IID conditions (CIFAR-100, $s = 5$ classes per client) after 100 communication rounds.

| Method | Accuracy (%) |
|---|---|
| FedAvg | 85.6 |
| FedAvg-M | 89.1* |
| FwdLLM | 3.5 |
| DeComFL | 4.4 |
| **FineFed** | **87.6** |

## J    FINEFED VS. FINEFED-HP: COMMUNICATION–ACCURACY–COMPUTE TRADE-OFFS

We compare FineFed-HT (forward-only head-gradient computation) and FineFed-HP (head perturbation) under matched settings. Unless otherwise noted, both variants keep the following components identical: (i) uncertainty-guided sample selection and micro-batch perturbations to estimate forward gradients for $\theta_{\text{PEFT}}$, and (ii) local model update with shared momentum. The sole difference is the head update mechanism: FineFed-HP replaces forward-only head tuning with perturbation-based estimation at the head. Similar to DeComFL, the model update and the shared momentum state in FineFed-HP can be reconstructed from seeds and projections. Experiments use ViT-B/16 with BitFit, with $P=8$ perturbations for $\theta_{\text{PEFT}}$.

FineFed-HT communicates at MB scale and requires only $P_{\text{head}}=1$ (single head forward). FineFed-HP achieves KB-level communication by reconstructing head perturbations with seeds and projections. To attain accuracy comparable to HT, FineFed-HP requires large number of head-perturbations:

Table 15: Quantitative trade-offs (accuracy %, communication per round, and head perturbations).

| Method | CIFAR-10 | CIFAR-100 | TinyImageNet | Comm./round | $P_{\text{head}}$ |
|---|---|---|---|---|---|
| FineFed(main, HT) | 97.5 | 88.0 | 84.9 | MB-scale | 1 |
| FineFed-HP | 97.1 | 83.1 | 75.8 | KB-scale | 128 |
| FineFed-HP | 97.5 | 86.5 | 81.5 | KB-scale | 1024 |
| FineFed-HP | – | 87.7 | 83.1 | KB-scale | 4096 |

$P_{\text{head}}$=1024 matches HT on CIFAR-10 (97.5) and reaches 86.5 on CIFAR-100; increasing to $P_{\text{head}}$=4096 further narrows the gap (87.7 on CIFAR-100; 83.1 on TinyImageNet) but still remains below HT (88.0/84.9). This illustrates that KB-level communication comes at the cost of substantial client-side computation due to excessive head perturbations.

In addition, FineFed-HP cannot be directly integrated into backward-based FL frameworks or hybrid settings that rely on backward-capable updates, since such updates cannot be reconstructed from seeds/projections. Overall, this quantifies the trade-off: reducing communication entails higher client computation and modest accuracy degradation; in practice, one can choose the method that best matches the real deployment constraints and needs.

## K  RELATED WORKS

**Zeroth-Order Optimization in Federated Learning.** FedZO (Fang et al., 2022) develops communication-efficient stochastic ZO methods for federated optimization. FwdLLM (Xu et al., 2024) adopts symmetric two-point ZO with centralized, variance-gated updates (server-centric stability, slower updates). DeComFL (Li et al., 2024b) realizes KB-level communication via seed/projection exchange (communication efficiency, higher perturbation and reconstruction cost). Contemporaneous with our work, FedMeZO (Ling et al., 2024) provides a theoretical account of ZO federated tuning for LLMs, establishing convergence rates (e.g., $\mathcal{O}\big(r^{3/2}(NKT)^{-1/2}\big)$ in IID) and clarifying the roles of learning rate ($\sim 1/\sqrt{d}$), local iterations $K$, client number $N$, perturbation scale $\mu$, and heterogeneity. In an orthogonal, systems-oriented line, FineFed addresses practical inefficiencies in existing forward-training frameworks—many-class optimization, perturbation budgeting, and non-IID stability—and further enables reliable optimization in many-class classification settings for robust real-world deployment.

**Centralized Zeroth-Order Optimization.** MeZO (Malladi et al., 2023), HELENE (Zhao et al., 2024), and HiZOO (Zhao et al., 2025) are centralized zeroth-order optimization algorithms for large language models. In principle, such centralized optimizers can be layered atop FineFed with appropriate design. We instantiate this by integrating HiZOO (*FineFed-HiZOO*). On ViT-B/16 with BitFit, FineFed-HiZOO attains 97.0/85.5/81.3 on CIFAR-10/100/TinyImageNet, whereas FineFed achieves 97.5/88.0/84.9. Thus, while feasible, the HiZOO integration does not yet improve convergence speed or final accuracy under distributed, non-IID settings, suggesting further algorithmic adaptation is needed.

**The Forward-Forward Algorithm** (FF) (Hinton, 2022) takes two forward passes for training neural networks. This layer-wise greedy learning method eliminates back-propagation by training each layer independently using a measure of "goodness".

Goodness is defined as the squared sum of activation for a given layer. During training, each layer undergoes two phases: a "positive" processing real data and a "negative" with generated negative samples. The objective is to maximize the goodness of positive exceeding a threshold $\theta$ while minimizing the goodness of negative below this threshold.

**FedFwd** (Park et al., 2023) applies the FF algorithm to FL. On MINST, it achieves 96.78% test accuracy with three hidden layers under IID settings and 45.76% accuracy on CIFAR-10, compared to FedAvg's 98.25% and 59.91%. While FedFwd reduces memory usage, it shows slower convergence in non-IID settings. The method remains untested on larger datasets.

## L  FineFed-HiZOO: Integration with Advanced Zeroth-Order Optimizers

To explore the potential of integrating advanced zeroth-order optimization methods with *FineFed*, we develop *FineFed-HiZOO* on top of *FineFed* by incorporating HiZOO (Zhao et al., 2025). Since HiZOO accelerates optimization using second-order (Hessian-based) information rather than momentum, the variant differs from *FineFed* only in replacing the shared momentum with a shared Hessian state aggregated across clients to couple updates across rounds.

### L.1  Experimental Setup

We conducted comprehensive experiments on ViT-B/16 with BitFit across CIFAR-10, CIFAR-100, and TinyImageNet datasets. The hyperparameter configuration for *FineFed-HiZOO* was determined through grid search:

Table 16: Hyperparameter configuration for FineFed-HiZOO

| Parameter | Value |
|---|---|
| Hessian smooth factor | 0.001 |
| Learning rate | 0.001 |
| Batch size | 64 |
| Number of perturbations ($P$) | 8 |
| Smooth factor $h$ (HiZOO $\mu$) | 0.001 |

### L.2  Results and Analysis

Table 17 presents the convergence results of *FineFed-HiZOO* across different datasets, showing the test accuracy achieved at various communication rounds.

Table 17: FineFed-HiZOO convergence results: Test accuracy (%) across communication rounds

| Round | CIFAR-10 | CIFAR-100 | TinyImageNet |
|---|---|---|---|
| 0 | 66.7 | 16.3 | 21.3 |
| 5 | 93.6 | 61.5 | 55.6 |
| 10 | 94.8 | 67.6 | 68.8 |
| 15 | 95.6 | 73.9 | 72.9 |
| 20 | 96.1 | 76.5 | 76.1 |
| 25 | 96.3 | 79.4 | 77.7 |
| 30 | 96.4 | 81.4 | 78.9 |
| 35 | 96.7 | 83.0 | 79.3 |
| 40 | 96.3 | 84.0 | 79.9 |
| 45 | 96.5 | 84.1 | 79.5 |
| 50 | 96.8 | 84.5 | 79.9 |
| 55 | 96.8 | 84.9 | 80.2 |
| 60 | 96.7 | 84.9 | 80.0 |
| 65 | 97.0 | 84.9 | 80.7 |
| 70 | 96.9 | 85.2 | 80.6 |
| 75 | 97.0 | 85.1 | 80.9 |
| 80 | 96.9 | 85.6 | 81.0 |
| 85 | 97.0 | 85.6 | 80.8 |
| 90 | 97.1 | 85.6 | 81.1 |
| 95 | 96.9 | 85.6 | 81.3 |
| 99 | 97.0 | 85.5 | 81.3 |

Table 18 compares the final performance of *FineFed-HiZOO* with the standard *FineFed* approach.

Table 18: Performance comparison: FineFed vs. FineFed-HiZOO (best test accuracy %)

| Method | CIFAR-10 | CIFAR-100 | TinyImageNet |
|---|---|---|---|
| FineFed-HiZOO | 97.0 | 85.5 | 81.3 |
| **FineFed** | **97.5** | **88.0** | **84.9** |

## L.3 KEY FINDINGS

Our experiments reveal several important insights:

- **Competitive but not superior performance:** While *FineFed-HiZOO* achieves competitive results, it does not outperform the standard *FineFed* approach in either convergence speed or final accuracy.

- **Distributed setting challenges:** The integration of HiZOO's second-order information in federated settings presents unique challenges that are not present in centralized optimization scenarios.

- **Hyperparameter sensitivity:** The performance of *FineFed-HiZOO* is highly sensitive to hyperparameter choices, particularly the Hessian smooth factor and learning rate.

These results suggest that directly transferring centralized zeroth-order optimization methods to federated learning may require further algorithmic adaptation and theoretical analysis to yield consistent improvements over simpler approaches like *FineFed*.

## M LIMITATIONS

While *FineFed* achieves strong performance and introduces a practical and efficient forward training framework for federated fine-tuning, we acknowledge several limitations that present opportunities for future research:

- **Accuracy–efficiency trade-offs.** Although *FineFed* significantly improves efficiency over existing forward training baselines, it may still underperform full backpropagation methods in highly complex tasks or settings that require deeper adaptation. This trade-off reflects the inherent balance between resource efficiency and fine-tuning capacity in constrained environments.

- **Hyperparameter sensitivity.** Certain components of *FineFed*, such as uncertainty-guided sample selection and micro-batch accumulation, may require careful tuning to achieve optimal performance across different datasets and model architectures. Future work could explore adaptive or automated tuning strategies to improve robustness.

- **Limited evaluation on NLP tasks.** While this work includes experiments on language classification benchmarks, we leave the evaluation of *FineFed* on generative language tasks—where output space and loss dynamics differ significantly—as promising future work.

- **Scope of theoretical guarantees.** Although this paper does not include new convergence proofs for the entire framework, several components of *FineFed*—such as the use of shared momentum—are grounded in prior theoretical results, which we cite. A formal convergence analysis of the full system under federated settings remains an open research direction.

These limitations do not undermine the core contributions of *FineFed*, but rather highlight directions to further extend its applicability and theoretical grounding.

## N    USE OF LARGE LANGUAGE MODELS (LLMS)

We used large language models (LLMs) solely for light language editing (e.g., grammar and wording) and for suggesting alternative phrasings. No ideas, methods, analyses, experiments, or citations were generated by LLMs. All technical content and conclusions are authored and verified by the authors, and no private or sensitive data were shared with LLM tools.

