# OpenReview forum: "FineFed: Forward-Only Federated Fine-Tuning for Many-Class Tasks under Non-IID Heterogeneity"
_ICLR.cc/2026/Conference — Submitted to ICLR 2026_

### Official Review · Reviewer_ioTB · 2025-10-18

**Soundness:** 2
**Presentation:** 2
**Contribution:** 3
**Rating:** 4
**Confidence:** 4

**Summary:**

This paper proposes FineFed, a forward-only FL framework addressing the challenge of fine-tuning large transformer models on resource-constrained edge devices. Based on the dependence of existing methods on memory-intensive backpropagation, FineFed utilizes forward-only head tuning and uncertainty-guided gradient estimation to realize efficient fine-tuning with less computational costs. Besides, FineFed employs a momentum-sharing mechanism to overcome the non-converging issue of FL over non-iid data.

**Strengths:**

1. FineFed outperforms the baselines in the experiment.

2. This paper has good reproducibility.

**Weaknesses:**

1. This paper should not put the literature review in the appendix.

2. The design of momentum sharing is not new. Similar ideas have already been proposed in [1-3].

3. Uncertainty-based sample selection (section 3.2) employs a batch-wise sampling stragety. In this scenario, if the batches in a dataset are not will-shifted due to a bad choice of random seed, its efficacy may be compromised by unbalanced sample batches.

4. The momentum sharing mechanism is orthogonal to forward-only head tuning and uncertainty-guided gradient estimation, and can be applied to other baselines as well.

5. Aside from model parameters, transmitting additional momentum may cause extra communication costs.

6. The presentation of this paper is not very clear.

**Questions:**

1. The presentation of this paper needs to be improved. For instance, concepts like “KB-level” communication (page 3) needs to be explained more clearly. Also, in lines 253-254, the authors state selecting 25% of total batches is sufficient. To support this claim, the authors should add a reference pointing at the corresponding empirical result. Lastlty, it is unclear how FineFed-HP works, and the difference between FineFed and FineFed-HP should be justified.

2. It seems that the forward gradient estimation (eq2) requires a closed-form representation of the gradient. Although such a representation is available for head layers with matrix-like shapes, for the complex feature extractors (e.g. convolutional layers), computing the closed-form representation of gradient is usually prohibitive.

3. FineFed is sensitive to the number of perturbation directions $P$ and the number of uncertain samples $k$. The authors should run some hyperparameter tuning experiments to show the effect of these hyperparameters.

4. The authors should add FedMeZO (discussed in the literature review) to the experimental baselines.

5. As shown in Algorithm 1 (lines 244-245), FineFed iteratively accumulates the gradient of PEFT, and I’m curious how to prevent the gradient-exposion problem in this scenario.

6. The authors need to specifically explain which parameters are PEFT parameters, and how to determine these parameters.

7. For fairness, the authors are supposed to evaluate the baselines with momentum sharing as well.


[1]On the Role of Server Momentum in Federated Learning, AAAI 2024.

[2]Momentum Benefits Non-IID Federated Learning Simply and Provably, ICLR 2024.

[3]Accelerating Federated Learning via Momentum Gradient Descent, IEEE Transactions on Parallel and Distributed Systems, 2020.

---

> ### Author Response · Authors · 2025-11-28
> **Response to Reviewer ioTB**
>
> We appreciate your positive assessment of FineFed's performance and reproducibility. We address your questions below.
>
> ### **W1. Literature review placement**
> We will move the Related Work section back to the main text in the final version.
>
> ### **W2. Novelty of Shared Momentum**
> While Server Momentum is a known technique, its application in **Zeroth-Order-Optimization in FL (ZOO-FL)** is non-trivial and critical.
> *   **Novel Insight:** ZOO-FL suffers from a "double variance" problem: **Gradient Noise** (from ZO) + **Data Heterogeneity** (from FL).
> *   **Unique Role:** We leverage Cheng et al. (ICLR 2024) to show that Shared Momentum provides a **Dual Benefit**: it (1) **Decouples Heterogeneity** by eliminating the convergence dependency on $\zeta$ (limiting baselines like DeComFL), and (2) **Reduces Variance** by dampening the ZO noise by $\frac{1-\beta}{1+\beta}$. This is essential for survival in the high-noise ZOO-FL regime. Our Ablation Study (Sec 4.4) confirms that without it, ZOO-FL suffers significant performance degradation in Non-IID settings.
>
> ### **W3. Uncertainty Sampling Strategy**
> Unbalanced batches may affect mini-batch methods, but our design mitigates this:
> 1.  **Scope:** Uncertainty sampling is **only** used for the gradient estimation phase; the classification head fine-tuning uses the **entire batch**.
> 2.  **Smoothing:** **Shared Momentum** effectively smooths out the sampling noise from single iterations, mitigating the impact of unbalanced batches.
>
> ### **W4. Orthogonality**
> We agree Shared Momentum is orthogonal to Forward-Only Head Tuning and Uncertainty-Guided Gradient Estimation. To demonstrate this, we conducted a **Shared Momentum Ablation Study (Appendix H)** comparing:
> 1.  **FwdLLM:** Uses its original *Cosine Similarity based Perturbation Sampling*.
> 2.  **FineFed (Momentum Only):** Equivalent to **DeComFL + Momentum**.
> 3.  **FineFed:** The full method (Momentum + Head Tuning + Uncertainty).
>
> The results show that while adding Momentum to DeComFL (i.e., FineFed-Momentum Only) improves performance, the full **FineFed** significantly outperforms all baselines, proving the necessity of our holistic design.
>
> ### **W5. Communication Overhead**
> The **3 MB** cost covers **PEFT parameters** and the **Shared Momentum** vector.
> 1.  **Negligible Cost:** The total size (~3 MB) is trivial for modern networks (4G/5G/WiFi).
> 2.  **Critical Gains:** This marginal bandwidth investment ensures **Convergence Stability** on many-class tasks—where DeComFL fails (7.2% acc)—and enables **Hybrid Training** with backward-capable clients, further enhance accuracy (improving from 84.9% to 85.8%).
> 3.  **Trade-off:** FineFed prioritizes **Usability and Performance** over extreme bandwidth compression, offering a robust solution where others fail.
>
> ### **W6. Presentation Clarity**
> We will improve the presentation as requested, including clarifying "KB-level" definitions and forward-only terminology.
>
> ### **Q1. Presentation Clarifications**
> 1.  **"KB-level" Communication**: Refers to methods (FineFed-HP, DeComFL) transmitting scalars/seeds (\~2 KB), whereas FineFed transmits PEFT parameters and momentum (\~3 MB).
> 2.  **Sampling Sufficiency (25%)**: We justify this with our **Ablation Study (Table 4 in Sec 4.4)**. On TinyImageNet, using 25% samples achieves **84.9%** accuracy, comparable to **85.0%** with All Samples (100%), but requires **$4\times$ less computation**. We will add this cross-reference.
>
> ### **Q2. Complex Extractors**
> Forward-Only Head Tuning only requires **input activations** to the final classifier. It treats the backbone as a black box, making it compatible with any architecture (ViT, RoBERTa, CNNs).
>
> ### **Q3. Hyperparameters**
> Please see Appendix F for sensitivity analysis on Computation Budget (perturbations per sample $P$ $\times$ selected samples $k$).
>
> ### **Q4. FedMeZO Baseline**
> DeComFL (ICLR 2025) is the SOTA advancement over FedMeZO (KDD 2024). The core difference is significantly reduced communication overhead while FedMeZO requires transmitting PEFT parameters. Thus, comparing against DeComFL is sufficient.
>
> ### **Q5. Gradient Explosion**
> We apologize for the omission. We explicitly use **Gradient Clipping (Max Norm = 1)** to prevent explosion, and will add this to Algorithm 1 and the experimental setup.
>
> ### **Q6. PEFT Params**
> We clarify that PEFT parameter details (Target Modules, Rank Size, etc.) are in **Appendix C.2**, and exact counts of trainable parameters (Head + PEFT) are in **Table 6**. We will make these references more prominent.
>
> ### **Q7. Baselines + Shared Momentum**
> As noted in W4:
> 1.  **FwdLLM** already incorporates momentum-like behavior via Cosine Similarity-based Perturbation Sampling.
> 2.  **FineFed (Momentum Only)** is equivalent to **DeComFL + Momentum**.
> Our Ablation Study demonstrates that adding Momentum alone is insufficient; **Head Tuning and Uncertainty** are crucial. Thus, we have effectively evaluated the momentum-enhanced baselines.

---

### Official Review · Reviewer_9TiC · 2025-10-28

**Soundness:** 2
**Presentation:** 3
**Contribution:** 2
**Rating:** 2
**Confidence:** 4

**Summary:**

This paper addresses the challenge of forward-only training in federated learning under many-class and strongly non-IID settings. It identifies that existing forward-only and zero-order gradient methods often suffer from unstable convergence and reduced accuracy when fine-tuning classification heads for heterogeneous clients. The proposed FineFed framework introduces a forward-only head tuning method, an uncertainty-guided forward gradient mechanism, and a shared momentum aggregation strategy. Experiments on NLP and vision benchmarks, combined with parameter-efficient tuning methods including BitFit, Adapter, and LoRA, demonstrate that FineFed’s effectiveness.

**Strengths:**

1.	While each component is individually known, the specific combination (forward-only analytical head gradients, uncertainty-guided perturbations, shared momentum) creates a robust and practical framework.
2.	Evaluations span multiple domains, tasks, and PEFT types, confirming generality and scalability.

**Weaknesses:**

1.	The proposed “FORWARD-ONLY HEAD TUNING FOR MANY-CLASS TASKS” is simple yet seems effective on large datasets such as CIFAR100 and ImageNet. However, it’s not clear why the proposed method converges well on these datasets, but the baselines such as FwdLLM and DeComFL cannot. More discussions and analysis are necessary to improve the theoretical depth.
2.	Despite the success on combining three key modules, the paper lacks convergence or variance-bound analysis for the combined use of single-sided forward gradients and shared momentum.
3.	The main FineFed model still requires MB-level uplink/downlink bandwidth, comparable to FedAvg-M. Competing forward-only methods (FwdLLM, DeComFL) use KB-level communication. Although FineFed-HP (head perturbation) achieves lower communication, it doubles computation cost and slightly reduces accuracy.
4.	Experiments use ≤100 clients with fixed-class (pathological) non-IID partitions. It remains unclear whether FineFed scales to thousands of clients or more realistic Dirichlet partitions.
5.	More experimental setups such as random seeds should be consolidated in the appendix for easier reproducibility.

**Questions:**

Please see the weaknesses.

---

> ### Author Response · Authors · 2025-11-28
> **Response to Reviewer 9TiC**
>
> We thank you for your constructive feedback and for acknowledging FineFed as a "robust and practical framework." Below we address your concerns.
>
> ### **W1. Theory for FineFed’s Convergence on Many-Class Tasks**
> The convergence failure of baselines (FwdLLM/DeComFL) on many-class tasks stems from the **"Curse of Dimensionality" in Zeroth-Order (ZO) gradient estimation**. We analyze this using the standard **variance bound** (Nesterov & Spokoiny, 2017; Liu et al., 2018). Let $d = d_{\text{PEFT}} + d_{\text{head}}$, where $d_{\text{head}} = C \times d_f$.
>
> 1.  **ZO Baselines (Variance Explosion):** Variance scales linearly with total dimension:
> $$
> \mathbb{E}\big[\lVert \hat{\nabla}\_{\text{ZO}} - \nabla \rVert^2\big]
> \propto (d\_{\text{PEFT}} + C \cdot d\_f)\,\lVert \nabla \rVert^2 + \sigma\_{\text{ZO}}^2
> $$
>     As $C$ increases, high variance overwhelms the gradient signal, leading to slow convergence or divergence.
>
> 2.  **FineFed (Dimension Decoupling):** We use **Forward-Only Head Tuning** (analytical gradient) for the head, and ZO estimation only for the fixed-dimension PEFT block. The total variance becomes:
>     $$ \text{Var}\_{\text{FineFed}}
>     \approx \underbrace{O(d\_{\text
>     {PEFT}}) \|\nabla\_{\text{PEFT}}\|^2}_
>     {\text{ZO Variance}} +
>     \underbrace{\text{Var}\_{\text{SGD}}
>     (\text{head})}\_{\text{Analytic
>     Variance}} $$
>     By replacing the dimension-dependent term $O(C \cdot d_f)$ with standard SGD variance (independent of perturbation dimension), we effectively mitigate the **"Curse of Dimensionality"** in ZOO-FL.
>
> ### **W2. Theoretical Analysis of Shared Momentum**
> Based on Cheng et al. (ICLR 2024), we compare convergence rates of DeComFL and FineFed (momentum only):
>
> 1.  **Single-Sided Forward Gradients (DeComFL):** Depends explicitly on data heterogeneity $\zeta$:
>     $$ \mathcal{O}\left( \frac{\sqrt{d}}{\sqrt{NKPR}} + \frac{\zeta^{2/3}}{R^{2/3}} + \frac{1}{R} \right) $$
>     DeComFL requires a **sufficiently large number of communication rounds $R$** to suppress the heterogeneity term ($\propto R^{-2/3}$), making it communication-inefficient in Non-IID settings.
>
> 2.  **Single-Sided Forward Gradients + Shared Momentum (FineFed - Momentum Only):** **Independent** of $\zeta$:
>     $$ \mathcal{O}\left( \frac{\sqrt{\frac{1-\beta}{1+\beta}d}}{\sqrt{NKPR}} + \frac{1}{R} \right) $$
>     Shared Momentum provides a **dual benefit**: (1) Eliminates dependency on $\zeta$ (Decoupling Heterogeneity), and (2) Dampens ZO variance by $\frac{1-\beta}{1+\beta}$ (Variance Reduction). This ensures stable convergence even under high ZO noise and Non-IID conditions.
>
> ### **W3. Communication Budget vs. System Trade-offs**
> FineFed's higher communication cost (~3 MB vs DeComFL's ~2 KB) is a deliberate **system-level trade-off** for usability and performance (Appendix J):
>
> | Method | Comp. | Comm. | Mem. | Acc. (%) | Hybrid Training |
> | :--- | :--- | :--- | :--- | :--- | :--- |
> | **FedAvg-M** | 192 | 3 MB | 1136 MB | 87.2 | No |
> | **FwdLLM** | 640 | 1 MB | 405 MB | 7.1 | Possible |
> | **DeComFL** | 704 | **2 KB** | 405 MB | 7.2 | No |
> | **FineFed** | **192** | 3 MB | **406 MB** | **84.9** | **Yes (85.8%)** |
> | **FineFed-HP**| 384 | **2 KB** | **406 MB** | 83.1 | No |
>
> *   **Acceptable Communication Overhead:** ~3 MB (including shared momentum) is negligible in modern networks (4G/5G/WiFi) and fully practical.
> *   **Critical Gains:** This bandwidth secures **Convergence Stability** on many-class tasks and enables **Hybrid Training** with backward-capable clients.
> *   **Computation Efficiency:** FineFed reduces computational costs to ~1/3 of DeComFL and eliminates expensive client-side model reconstruction.
>
> ### **W4. Scalability & Partitioning**
> The **Pathological Partition with 100 clients** is the standard benchmark established by FedAvg paper (McMahan et al., 2017) for evaluating FL under Non-IID settings.
> We strictly follow this protocol to ensure a fair and direct comparison with baselines. This setting effectively creates the extreme statistical heterogeneity required to stress-test ZO-based methods, validating FineFed's robustness in challenging Non-IID scenarios.
>
> ### **W5. Reproducibility**
> We confirm that all experiments used fixed random seeds (Seed=0). We will consolidate all hyperparameter tables and hardware details into a unified Appendix section.

---

### Official Review · Reviewer_bNnK · 2025-10-29

**Soundness:** 2
**Presentation:** 2
**Contribution:** 2
**Rating:** 4
**Confidence:** 4

**Summary:**

This paper presents FineFed, a forward-only federated fine-tuning framework designed to make FL of large models feasible on resource-constrained clients. The authors aim to address the limitations of backpropagation-based fine-tuning and previous forward-only methods.

**Strengths:**

++The paper targets an important problem, i.e., making large model fine-tuning feasible on memory-constrained devices, which is a key bottleneck for future FL deployments.
++The description of forward-only head tuning is clear and rigorous.
++The paper presents diverse experiments across multiple domains, datasets, and PEFT strategies. Ablation studies isolate the impact of each component.
++The inclusion of code availability, hyperparameters, and detailed appendices improves reproducibility.

**Weaknesses:**

++The paper lacks any formal convergence or variance analysis for FineFed’s forward-gradient estimators or shared momentum dynamics.
++While combining the three techniques is novel in integration, each component is built on existing concepts with limited theoretical innovation. Therefore, the novelty is incremental.
++The accuracy-communication-compute trade-offs between FineFed and FineFed-HP are not fully analyzed under larger model scales.

**Questions:**

Please address all the weaknesses above.

---

> ### Author Response · Authors · 2025-11-28
> **Response to Reviewer bNnK**
>
> Thank you for recognizing FineFed's relevance to "memory-constrained devices" and our rigorous description. We address your concerns below.
>
> ### **W1. Missing formal convergence or variance analysis**
> We ground FineFed in the **variance bound of the gradient estimator** (Nesterov & Spokoiny, 2017) and momentum theory (Cheng et al., ICLR 2024).
>
> 1.  **Why ZO Baselines Fail ("Curse of Dimensionality"):** The variance of ZO gradient estimators scales with the parameter dimension. For baselines perturbing the high-dimensional head ($d_{\text{head}} \propto C$), the variance explodes as the number of classes $C$ increases ($ \text{Var} \propto C \cdot \|\nabla \mathcal{L}\|^2$). This overwhelming noise prevents convergence.
> 2.  **Why FineFed Converges:**
>     *   **Head Tuning:** Computes **exact analytical gradients** for the head. This **eliminates the dimension-dependent ZO variance** for the head parameters. The remaining variance comes from the low-dimensional PEFT block (ZO) and standard SGD noise (Analytical), both of which are **independent of the class count $C$**.
>     *   **Shared Momentum:** As proven by Cheng et al., momentum removes the dependence of convergence rate on data heterogeneity $\zeta$, stabilizing the updates in Non-IID settings.
>
> We will explicitly connect these theoretical foundations to our method in the revision to provide the requested depth.
>
> ### **W2. Perceived incremental novelty**
> FineFed is not a mere combination of components but introduces **three targeted algorithmic innovations** to overcome fundamental bottlenecks preventing Forward-Only FL on complex tasks:
>
> 1.  **Solving the "Curse of Dimensionality" in Many-Class Tasks:**
>     Existing ZO methods fail due to the **"Curse of Dimensionality"**, where variance explodes linearly with classes $C$. FineFed addresses this critical bottleneck by reducing the ZO gradient variance through two strategies: (1) **Forward-Only Head Tuning** (standard FineFed), which computes **exact analytical gradients** for the head to mathematically eliminate the dimension-dependent variance; or (2) **Increased Perturbations** (FineFed-HP), which suppresses variance through higher sampling density. This targeted variance reduction ensures convergence on many-class tasks where baselines diverge.
>
> 2.  **Uncertainty-Guided Gradient Estimation (Solving the Compute Bottleneck):**
>     A major barrier for ZO on edge devices is the high cost of perturbations required for accurate gradient estimation. We propose a novel **uncertainty-based sampling strategy** that dynamically prioritizes informative samples for perturbation. Unlike uniform sampling, this method efficiently allocates the compute budget, reducing the gradient estimation cost by **$\approx 2.5\times$** while maintaining accuracy.
>
> 3.  **Shared Momentum (Solving the Non-IID Stability Bottleneck):**
>     ZO gradients are inherently noisy. We show that **Shared Momentum** provides a dual theoretical benefit: it not only **decouples the convergence rate from data heterogeneity** (removing the $\zeta$ dependency), but also **suppresses the ZO gradient variance** by a factor of $\frac{1-\beta}{1+\beta}$ (effectively dampening noise by $\sim 19\times$ at $\beta=0.9$). This transforms the erratic updates of standard ZO into a stable trajectory, ensuring robustness even under extreme Non-IID partitions.
>
> **Summary:** FineFed’s contribution is a **cohesive framework** that solves the specific structural flaws (Variance, Cost, Instability) of prior Forward-Only FL methods, enabling the first practical deployment of large models on constrained edge devices.
>
> ### **W3. Insufficient trade-off analysis between FineFed and FineFed-HP**
> We summarize the trade-off analysis (**Section 4.3 & Appendix J**):
>
> | Method | Comp. | Comm. | Mem. | Acc. (%) | Hybrid Training |
> | :--- | :--- | :--- | :--- | :--- | :--- |
> | **FedAvg-M** | 192 | 3 MB | 1136 MB | 87.2 | No |
> | **FwdLLM** | 640 | 1 MB | 405 MB | 7.1 | Possible |
> | **DeComFL** | 704 | **2 KB** | 405 MB | 7.2 | No |
> | **FineFed** | **192** | 3 MB | **406 MB** | **84.9** | **Yes (85.8%)** |
> | **FineFed-HP**| 384 | **2 KB** | **406 MB** | 83.1 | No |
>
> *   **FineFed (Recommended):** The optimal balance. It achieves **superior accuracy (84.9%)** with minimal computation (matching FedAvg-M) while saving $\approx 64\\%$ memory. Its 3MB communication cost is negligible for modern networks and enables **Hybrid Training**, allowing the system to leverage clients with backward-propagation capabilities to further enhance accuracy (improving from 84.9% to 85.8%).
> *   **FineFed-HP (Bandwidth-Constrained):** A specialized variant for **extreme bandwidth limits**. It achieves ultra-low communication (2 KB) but incurs **$2\times$ computation cost** and slight accuracy degradation.
>
> **Conclusion:** As analyzed in the paper, FineFed is the robust default choice for memory-constrained many-class tasks. FineFed-HP is a specialized fallback strictly for bandwidth-critical scenarios.

---

### Official Review · Reviewer_WTnz · 2025-11-01

**Soundness:** 2
**Presentation:** 2
**Contribution:** 2
**Rating:** 4
**Confidence:** 2

**Summary:**

This paper aims to improve the performance of forward-only (zeroth-order) in federated learning. The authors propose FineFed, a hybrid framework that is based on three primary components: 1) Forward-Only Head Tuning (FineFed-HT), 2) Uncertainty-Guided Forward Gradient Estimation and 3) Shared Momentum.

The paper combines the memory-saving benefits of ZO for the large backbone with the fast-convergence of exact analytic gradients for the small but critical head. The empirical results show that FineFed outperforms prior forward-only baselines on many-class tasks and achieves accuracy comparable to backpropagation-based methods such as FedAvg-M.

**Strengths:**

1- The authors have clearly explained the existing methods in forward only methods in FL and their shortcomings.

2- Focusing on Head tuning especially for multi-class datasets is interesting.

3- The experiments are comprehensive and demonstrate the method's effectiveness.

4- The ablation study shows the importance of different design choices.

**Weaknesses:**

1- The authors mention that this is a forward-only (zeroth_order) paper but the "Forward-Only Head Tuning"  is the exact analytic first-order gradient of the cross-entropy loss with respect to the head parameters.

2- Authors explain that one of the benefits of forward-only methods is their communication efficiency. However this method requires MB-scale communication (Table 2), as it must transmit the full PEFT parameters and momentum state, just like FedAvg-M.

3- Overstated Novelty of Components: Some choices such as Shared Momentum or Uncertainty-Guided Sampling are very common and well-established techniques in machine learning to improve the performance or make the training more efficient.

4- The scope of the paper is limited to classification tasks.

**Questions:**

Please check out the previous section.

---

> ### Author Response · Authors · 2025-11-28
> **Response to Reviewer WTnz**
>
> Thank you for your review. We address your concerns regarding definitions and novelty.
>
> ### **W1. Definition of "Forward-Only"**
> Our definition aligns with the core system constraint: **No Backpropagation**.
> *   **System Bottleneck:** The main memory cost in training is **Activation Caching** required for auto-differentiation.
> *   **FineFed:** By calculating head gradients analytically using only forward activations, we **avoid constructing a computational graph** and caching intermediate activations. Thus, we retain the memory benefits of Forward-Only training while gaining the precision of First-Order methods.
>
> ### **W2. Communication Efficiency**
> We clarify that Zero-Order (ZO) methods like DeComFL offer two distinct benefits: **low memory footprint** and **low communication**, but suffering **high computation cost**. We provide a flexible framework to address both:
>
> *   **Extreme Bandwidth Constraints:** For scenarios where communication is the bottleneck, **FineFed-HP** aligns with ZO methods (sending only seeds/scalars ~2KB) while significantly outperforming them in stability and accuracy.
> *   **Standard Modern Networks:** In most edge scenarios, **3 MB** is a negligible cost. Here, standard **FineFed** is the superior choice. By utilizing this small bandwidth margin, FineFed achieves **more stable convergence, reduced computation ($2\times$ less than FineFed-HP), and higher accuracy**, making it the practical "sweet spot."
>
> | Method | Comp. | Comm. | Mem. | Acc. (%) | Hybrid Training |
> | :--- | :--- | :--- | :--- | :--- | :--- |
> | **FedAvg-M** | 192 | 3 MB | 1136 MB | 87.2 | No |
> | **FwdLLM** | 640 | 1 MB | 405 MB | 7.1 | Possible |
> | **DeComFL** | 704 | **2 KB** | 405 MB | 7.2 | No |
> | **FineFed** | **192** | 3 MB | **406 MB** | **84.9** | **Yes (85.8%)** |
> | **FineFed-HP**| 384 | **2 KB** | **406 MB** | 83.1 | No |
>
> **Summary:** We offer a versatile solution—**FineFed-HP** for extreme communication constraints (superior to DeComFL) and **FineFed** for balanced scenarios (superior to FedAvg-M in memory  and ZO methods in accuracy/compute). This flexibility is a key advantage of our framework.
>
> ### **W3. "Overstated Novelty" & Design Choice**
> We respectfully clarify that FineFed is not a mere combination of existing components, but a **cohesive framework** designed to solve the specific structural flaws of prior Forward-Only FL methods. As detailed in our response to other reviewers, each component targets a fundamental bottleneck:
>
> 1.  **Solving the "Curse of Dimensionality" (Head Tuning):** Existing ZO methods fail on many-class tasks because gradient variance explodes with the number of classes. **Forward-Only Head Tuning** computes exact analytical gradients for the head, mathematically eliminating this dimension-dependent variance.
> 2.  **Solving the "Compute Bottleneck" (Uncertainty Sampling):** Standard ZO requires excessive perturbations. Our **Uncertainty-Guided Sampling** reduces the compute budget by **$\approx 2.5\times$** by prioritizing informative samples.
> 3.  **Solving the "Non-IID Instability" (Shared Momentum):** ZOO-FL suffers from "double variance" (ZO noise + Data Heterogeneity). **Shared Momentum** provides a dual benefit: it decouples convergence from data heterogeneity (removing $\zeta$ dependency) and dampens ZO noise.
>
> **Conclusion:** This is not a random assembly of techniques, but a **Minimal Necessary Design** to make Forward-Only FL viable on edge devices. Each component is indispensable for addressing specific failure modes (Variance, Cost, Instability) of prior methods, as empirically verified in our Ablation Study (Sec 4.4 and Appendix H).
>
> ### **W4. Scope limited to classification tasks**
> *   **Current coverage:** FineFed already spans both NLP (SST-2, AGNews, SuperGLUE) and vision (CIFAR-10/100, TinyImageNet) tasks.
> *   **Future extensions:** While currently focused on classification (the dominant edge application), the **Forward-Only Fine-Tuning Framework** is generalizable. Extending to generative tasks is a promising future direction.

---

### Meta-Review · Area_Chair_m6ru · 2025-12-14

**Summary:**

In this submission, the authors focused on the problem of non-iid FL. While all reviewers acknowledged that the focused problem is important and the effectiveness of the proposed method, significant concerns were raised regarding theoretical analysis and presentation. Specifically, one reviewers raised questions the "forward-only" may not be proper, as Forward-Only Head Tuning" is the exact analytic first-order gradient of the cross-entropy loss.

**Reviewer Concerns:**

The AC read through the submission and rebuttal carefully, and found that while the comments regarding experiments are well addressed. The theoretical study, especially convergence, and the concern of "first-order gradient" are not well addressed. The authors failed to provide convergence bound, but only analysis. Also, the AC also concurs with the concerns that "forward only" is misleading.

**Reviewer Scores:**

The reviewers are less likely to change the scores.

---

### Decision · Program_Chairs · 2026-01-26

Reject